# New Strains of the Entomopathogenic Nematodes *Steinernema scarabaei*, *S. glaseri*, and *S. cubanum* for White Grub Management

**DOI:** 10.3390/insects15121022

**Published:** 2024-12-23

**Authors:** Albrecht M. Koppenhöfer, Ana Luiza Sousa

**Affiliations:** Department of Entomology, Rutgers University, 96 Lipman Dr., New Brunswick, NJ 08901, USA; sousa.alvs@gmail.com

**Keywords:** *Anomala orientalis*, *Cyclocephala borealis*, *Popillia japonica*, scarab, insect-parasitic nematodes, biological control

## Abstract

White grubs are important pests of many agricultural and horticultural crops. We tested four entomopathogenic nematode isolates recently isolated from infected white grubs in turfgrass areas in central New Jersey, USA for the biological control of three white grub species that are pests of turfgrass and ornamental plants. Against oriental beetle and Japanese beetle larvae, the *Steinernema scarabaei* Ad and *S. scarabaei* SL isolates caused very high mortality at low rates in laboratory tests, whereas *S. cubanum* HF was less virulent and *S. glaseri* SH the least virulent isolate. None of the isolates caused high mortality of northern masked chafer larvae. Nematode efficacy in greenhouse tests against oriental beetle larvae followed the same pattern. The original isolate of *S. scarabaei*, the AMK001 strain, that had been maintained in the laboratory on white grubs as hosts for 19 years, showed the same virulence level as it did soon after its first isolation and was also as virulent as the fresh *S. scarabaei* Ad isolate. Future tests should determine the ability of these white grub-adapted nematode isolates to provide long-term suppression of white grub populations.

## 1. Introduction

Entomopathogenic nematodes (EPNs) (Rhabditida: Heterorhabditidae and Steinernematidae) in tandem with their symbiotic bacteria (Enterobacterales: Morganellaceae) offer a safe and sustainable option for the management of a wide range of insect pests [1,2]. As the soil and soil litter environment is their natural habitat, EPNs tend to be particularly promising for the control of soilborne insect pests [1,3]. The root-feeding larvae of scarab beetles (Coleoptera: Scarabaeidae), also called white grubs, are pestiferous to a wide range of agricultural and horticultural plants around the world and have received a great deal of attention as targets of EPNs [1,4]. However, white grubs, having coevolved in the soil with many different soil pathogens and parasites, possess various defense mechanisms including infrequent carbon dioxide output, defensive and evasive behaviors, sieve-plates over their spiracles, frequent defecation, a dense peritrophic membrane, and a strong immune response that make them relatively resistant to infection by EPNs [4].

Numerous studies have shown that different white grub species can differ significantly in their susceptibility to different EPN species [1]. Moreover, the relative virulence of different EPN species and even strains thereof varies greatly among white grub species (e.g., [5,6,7]). Not surprisingly, species and strains thereof that were found naturally infecting white grubs in the field tend to be among the most virulent and field-effective EPN species for white grub control [1]. However, species that have shown particularly high potential for white grub control have also tended to be more restricted in their host range, such as *Steinernema kushidai* Mamiya and *S. scarabaei* Stock & Koppenhöfer, which have low virulence to non-scarab insect species [8,9,10].

Many of the studies on EPNs for white grub management have been conducted in eastern North America, especially the northeastern USA, where a complex of white grub species are the most destructive and widespread pests of turfgrasses and ornamental plants and also damage various horticultural crops. Important species include the native masked chafers, *Cyclocephala* spp., and introduced species such as the Japanese beetle, *Popillia japonica* Newman, oriental beetle, *Anomala orientalis* Waterhouse, and European chafer, *Rhizotrogus majalis* (Razoumowsky) [11,12].

Under favorable conditions, well-adapted EPN species/strains such as *Heterorhabditis bacteriophora* Poinar can provide curative *P. japonica* control equal to that of standard insecticides [13,14]. However, other white grub species such as *A. orientalis*, *R. majalis*, Asiatic garden beetle [*Maladera castanea* (Arrow)], and May/June beetles (*Phyllophaga* spp.) appear to be more difficult to control [4,6,7]. The only species that has provided good control of these species is *S. scarabaei* [4,9]. Due to its close adaptation to white grubs as hosts, *S. scarabaei* is highly virulent to a wide range of white grub species [6,7] and reproduces very well in them [10]. This enables it to provide long-term suppression of white grub populations after a single application at very low rates [15]. It can be argued that other EPN species or strains regularly found infecting white grubs, and hence likely to be well-adapted to them as hosts, could have similar high potential for white grub management. Unfortunately, *S. scarabaei* has thus far escaped attempts to mass produce it in vitro, which limits its commercialization potential. However, other white grub-adapted EPN species and strains, even of *S. scarabaei*, may be easier to mass produce.

The objective of this study was to determine the potential as white grub control agents of several new EPN isolates that had been found regularly infecting white grubs in a recent survey in turfgrass areas with a history of white grub infestation in central New Jersey, USA. In addition, the potential of new isolates of *S. scarabaei* was compared to that of the original *S. scarabaei* isolate, which has been in laboratory culture since 2001 and was suspected to have lost some of its original virulence. To this end, laboratory and greenhouse studies were conducted.

## 2. Materials and Methods

### 2.1. General Methods

Third-instar larvae *A. orientalis*, *P. japonica*, and *Cyclocephala borealis* Arrow were collected in early October in turf areas at Rutgers Plant Science Research and Extension Farm (Adelphia, NJ, USA; 40°23′ N, 74°26′ E) and Rutgers Horticultural Farm No. 2 (North Brunswick, NJ, USA, 40°47′ N, 74°42′ E). The larvae were kept individually in the cells of 24-well plates in sandy loam at 15 °C for short-term storage and at 6 °C for long-term storage (2–10 wk.). Before use in experiments, the larvae were warmed up at room temperature (21–24 °C) for 24 h.

*Steinernema scarabaei* AMK001 strain (GenBank accession number AY172023) was isolated from infected *A. orientalis* and *P. japonica* third instars found in turfgrass areas at the Rutgers Plant Science Research and Extension Farm (Adelphia, NJ) in October 2000 and May 2001 [16]. It has been continuously maintained in culture through bi-yearly propagation in third instars of *A. orientalis* and *P. japonica* as it cannot be reliably reared in late instar larvae of the greater wax moth, *Galleria mellonella* (L.). Moreover, rearing it in its original host species infected in 30 mL cups with soil and annual ryegrass (*Lolium multiflorum* Lam.) as a food source for the larvae was meant to reduce the likelihood of lab adaptation and loss of white grub adaptation.

Fresh isolates of white grub-adapted EPNs were found in a survey of turfgrass areas with a history of white grub infestations in central New Jersey that was conducted in late September and early October of 2019. EPN-infected white grubs were rinsed with tap water, then 70% ethanol, and again tap water to remove soil and other contaminants from the cadaver cuticle. Then the cadavers were placed on emergence traps and any emerging infective juvenile (IJ) EPNs were collected in the water of the traps and stored in tissue culture flasks at 6 °C [17]. These IJs were then used to infect third-instar *A. orientalis* in 30 mL cups with soil and annual ryegrass, and the resulting cadavers were placed on emergence traps to collect and store emerging IJs as above. Samples of distinct isolates based on white grub cadaver color, and the size and behavior of the IJs, were sent for molecular and microscopic diagnostics to Dr. Patricia Stock’s laboratory at the University of Arizona (Tucson, AZ, USA). The nematodes were identified to species using the 18S, ITS-2, and 28S genes. The four distinct isolates were *S. scarabaei* Ad, isolated at the Rutgers University Plant Science Research and Extension Farm; *S. scarabaei* SL, isolated from Spring Lake Golf Club (Spring Lake, NJ, USA, 40°15′ N, 74°04′ E); *S. glaseri* Steiner SH, isolated from Shark River Golf Course (Neptune, NJ, USA, 40°20′ N, 74°06′ E); and *S. cubanum* Mracek HF, isolated from Rutgers Horticultural Farm No. 2.

To produce IJs for experiments, third-instar *A. orientalis* were exposed to about 100 IJs of a given EPN isolate in 30 mL cups with soil and annual ryegrass, the resulting cadavers placed on emergence traps, and any emerging IJs collected for 7–12 days and stored in tap water in a tissue culture flask at 6 °C for 7–21 days [17]. Before use in experiments, IJs were acclimatized for 6 h at room temperature (21–24 °C). The soil used in the experiments was a sandy loam (61% sand, 27% silt, 12% clay, 2.3% organic matter, pH 5.9) that had been pasteurized (3 h at 70 °C) and air-dried before use.

### 2.2. Laboratory Experiments

Laboratory experiments were conducted at room temperature (21–24 °C) in 30 mL plastic cups (10 cm^2^ surface area). Dry soil and annual ryegrass seed (1% *w*/*w* relative to soil) were mixed, water was mixed in to 11% (*w*/*w*) soil moisture, and 20 g of the mix filled in each cup. After 3 days to allow for grass germination, individual larvae were released into the cups. Larvae that did not enter the soil within 2 h were replaced. The cups were treated 1 day later. Treatments were applied in 0.5 mL water (final soil water potential −20 kPa = 13% *w*/*w* soil moisture). Controls received water only. There were 30 cups per treatment and for the untreated control in each experiment run. Larval mortality was assessed at 7 and 14 days after treatment (DAT). Nematode rate ranges were based on previous observations with different strains of the same species (e.g., [7,9]).

Experiments 1–3 compared the virulence of the four new EPN isolates to the three white grub species. In Experiment 1, third-instar *A. orientalis* were exposed to *S. scarabaei* Ad, *S. scarabaei* SL, *S. glaseri* SH, and *S. cubanum* HF, each at 25, 100, and 400 IJs/larva. The experiment was run three times for a total of 90 cups per treatment. In Experiment 2, third-instar *P. japonica* were exposed to the same treatments, and the experiment was run twice. In Experiment 3, third-instar *C. borealis* were exposed to 100 and 400 IJs/larva of the same EPN isolates, and the experiment was run twice; the 25 IJs/larva rate was not used since previous studies had suggested that *C. borealis* was less susceptible to EPNs than *A. orientalis* and *P. japonica* [1,4,6,7]. Experiment 4 compared the virulence of the two new *S. scarabaei* isolates to that of the original *S. scarabaei* AMK001 strain against third-instar *A. orientalis*. The larvae were exposed to 13, 25, and 50 IJs/larva, and the experiment was run twice.

### 2.3. Greenhouse Experiments

A greenhouse experiment was conducted to confirm the observations on the concentration responses under more field-like conditions. Plastic pots (10 cm × 10 cm × 12 cm; 100 cm^2^ at soil surface) were filled with sandy loam to a height of 10 cm and seeded with annual ryegrass. The pots were watered every 2–3 days until the end of the experiment and the grass cut weekly to a 5 cm height with scissors. Four weeks after the grass was sown and 1 day before treatment application, five third-instar *A. orientalis* were introduced per pot. Larvae that did not enter the soil within 6 h were replaced. Treatments were *S. scarabaei* Ad, *S. scarabaei* SL, *S. glaseri* SH, and *S. cubanum* HF, each applied at 313 and 625 IJs/pot (=0.313 and 0.625 × 10^9^ IJs/ha). Treatments were applied in 50 mL of water. Controls received 50 mL water only. After application, pots were arranged in a completely randomized design. The number of surviving larvae was determined at 14 DAT by destructively searching through the soil. The greenhouse was maintained at 28 °C/18 °C (day/night; 14/10 h L/D) and the soil temperature in the pots averaged 22.8 ± 1.6 °C. There were six pots and seven pots per treatment in the first and second experiment runs, respectively.

### 2.4. Statistical Analysis

In the laboratory experiments, groups of 10 cups were considered as replicates. Survival data per group of 10 cups in the laboratory and per pot in the greenhouse were transformed into percent mortality data, corrected for control mortality [18], and arcsine square root transformed before analysis using ANOVA (software Statistix 10.0; Analytical Software, 2018), with experiment run as a factor and means separation using Tukey’s test. To determine which treatments provided significant control compared to the untreated control in the greenhouse, survival data were square root transformed and analyzed with ANOVA, with experiment repetition as a factor followed by means separation using Tukey’s test. Differences among means were considered significant at *p* < 0.05. Means ± SEM are presented.

## 3. Results

### 3.1. Laboratory Experiments

In the first experiment, *A. orientalis* mortality in the untreated control was 1% at 7 DAT and 4% at 14 DAT. Control-corrected mortality followed the same pattern at 7 and 14 DAT. Mortality was significantly affected by EPN isolate (F_2, 107_ ≥ 66.98; df = 3, 107; *p* < 0.001), EPN rate (F_2, 107_ ≥ 66.98; *p* < 0.001), and experiment run (F_2, 107_ ≥ 5.55; *p* < 0.01); there were no significant interactions between any of the factors (*p* ≥ 0.07). Across all four isolates, mortality increased significantly from 25 IJs/larva to 100 IJs/larva to 400 IJs/larva. *Steinernema scarabaei* Ad caused the highest mortality overall at 7 and 14 DAT, but within rate caused significantly higher mortality than *S. scarabaei* SL only at 25 IJs/larva at 7 and 14 DAT (Figure 1A). Both *Steinernema scarabaei* isolates caused greater mortality than *S. cubanum* HF and *S. glaseri* SH at 7 and 14 DAT overall, and at all rates except for the 25 IJs/larva rate at 7 DAT, when *S. scarabaei* SL did not cause significantly higher mortality than *S. cubanum* HF. *Steinernema cubanum* HF caused significantly higher mortality than *S. glaseri* SH at 7 and 14 DAT overall and at all rates except for at 25 IJs/larva at 14 DAT (Figure 1A).

In the second experiment, *P. japonica* mortality in the untreated control was 4% at 7 DAT and 8% at 14 DAT. Control-corrected mortality followed a similar pattern at 7 and 14 DAT. Mortality was significantly affected by EPN isolate (F_2, 71_ ≥ 40.33; *p* < 0.001) and EPN rate (F_2, 71_ ≥ 61.79; *p* < 0.001) but not by experiment run (*p* ≥ 0.462); there was a significant interaction between EPN isolate and rate (*p* ≤ 0.05). At 7 DAT, both *S. scarabaei* isolates caused similar mortality, but only *S. scarabaei* Ad caused significantly higher mortality than *S. cubanum* HF at all rates, whereas *S. scarabaei* SL only caused significantly higher mortality than *S. cubanum* HF at 25 IJs/larva and 400 IJs/larva (Figure 1B). *Steinernema cubanum* HF caused significantly higher mortality than *S. glaseri* SH at 100 IJs/larva and 400 IJs/larva but not at 25 IJs/larva. At 14 DAT, both *S. scarabaei* isolates caused similar mortality, which was significantly higher than mortality caused by *S. cubanum* HF only at 25 and 100 IJs/larva (Figure 1B). *Steinernema cubanum* HF caused significantly higher mortality than *S. glaseri* SH at 100 and 400 IJs/larva but not at 25 IJs/larva.

In the third experiment, *C. borealis* mortality in the untreated control was 0% at 7 and 14 DAT. At 7 DAT, control-corrected mortality was significantly affected by EPN rate (F_1, 47_ = 14.94; *p* < 0.001) but not by EPN isolate (*p* = 0.074) and experiment run (*p* = 0.220); there were no significant interactions between any factors (*p* ≥ 0.07). *Steinernema scarabaei* Ad and *S. cubanum* HF caused greater mortality than *S. scarabaei* SL, whereas mortality caused by *S. glaseri* SH did not differ significantly from any other isolate. Across isolates, 400 IJs/larva caused greater mortality than 100 IJs/larva. Mortality at 7 DAT did not exceed 22% (Figure 1C). At 14 DAT, mortality was significantly affected by EPN isolate (F_3, 47_ = 6.45; *p* < 0.01), EPN rate (F_1, 47_ = 34.63; *p* < 0.001), and experiment run (F_1, 47_ = 7.71; *p* < 0.01); there were no significant interactions between any factors (*p* ≥ 0.42). *Steinernema scarabaei* Ad caused significantly greater mortality than *S. cubanum* HF and *S. scarabaei* SL but not *S. glaseri* SH. *S. glaseri* SH caused significantly greater mortality than *S. scarabaei* SL but not *S. cubanum* HF. However, even the high rate of *S. scarabaei* Ad only caused 72% at 14 DAT (Figure 1C).

In the fourth experiment, *A. orientalis* mortality in the untreated control was 6% at 7 DAT and 12% at 14 DAT. At 7 DAT, mortality was significantly affected by *S. scarabaei* isolate (F_2, 53_ = 50.02; *p* < 0.001), EPN rate (F_2, 53_ = 14.16; *p* < 0.001), and experiment run (F_1, 53_ = 13.59; *p* < 0.01); there were no significant interactions between any factors (*p* ≥ 0.08). Across isolates, 50 and 25 IJs/larva caused greater mortality than 13 IJs/larva. *Steinernema scarabaei* Ad and *S. scarabaei* AMK001 caused similar mortality, which was higher than that caused by *S. scarabaei* SL overall and at each rate (Figure 2). Both superior isolates caused at least 70% mortality at 13 IJs/larva and at least 83% at 50 IJs/larva (Figure 1C). At 14 DAT, mortality was significantly affected by EPN isolate (F_2, 53_ = 12.84; *p* < 0.001) and EPN rate (F_2, 53_ = 15.34; *p* < 0.001) but not experiment run (*p* = 0.091); there were no significant interactions between any factors (*p* ≥ 0.481). Across isolates, 50 IJs/larva caused greater mortality than 25 IJs/larva, which caused greater mortality than 13 IJs/larva. *Steinernema scarabaei* Ad and *S. scarabaei* AMK001 caused similar mortality, which was higher than that caused by *S. scarabaei* SL. *Steinernema scarabaei* AMK001 caused greater mortality than *S. scarabaei* SL at 50 and 13 IJs/larva whereas *S. scarabaei* Ad caused significantly higher mortality than *S. scarabaei* SL only at 13 IJs/larva (Figure 2). The superior isolates caused 78% mortality at 13 IJs/larva and 87–88% at 50 IJs/larva.

### 3.2. Greenhouse Experiment

In the untreated control, 97% of the released *A. orientalis* larvae were recovered alive. All treatments had significantly fewer alive larvae than the untreated control (F_8, 99_ = 37.63; *p* < 0.001) and there was no significant effect of experiment run (*p* = 0.563) and no interaction between treatment and experiment run (*p* = 0.097). Control-corrected percent mortality was significantly affected by EPN isolate (F_3, 103_ = 38.96; *p* < 0.001) and EPN rate (F_1, 103_ = 38.16; *p* < 0.001) but not by experiment run (*p* = 0.662); there were no significant interactions between any factors (*p* ≥ 0.07). The high EPN rate caused greater mortality than the low rate and both *S. scarabaei* isolates caused greater mortality than *S. cubanum* HF, which caused greater mortality than *S. glaseri* SH (Figure 3). At 0.625 × 10^9^ IJs/ha, the *S. scarabaei* isolates caused 77–83% mortality, whereas *S. cubanum* HF and *S. glaseri* SH only caused 49% and 42%, respectively, mortality.

## 4. Discussion

All four EPN isolates that we had originally isolated from infected white grubs in the field infected third-instar larvae of the three tested white grub species. Against *A. orientalis* and *P. japonica* larvae, both *S. scarabaei* isolates were highly virulent in laboratory tests, whereas *S. cubanum* HF caused only high mortality at a high IJ rate and *S. glaseri* SH did not even cause high mortality at the high IJ rate. None of the isolates caused high mortality of *C. borealis* larvae. EPN efficacy in the greenhouse tests followed a very similar pattern. The original isolate of *S. scarabaei*, the AMK001 strain, was found to have maintained its high level of virulence after 19 years of laboratory culture and had the same virulence level as the fresh *S. scarabaei* Ad isolate; both were more virulent than *S. scarabaei* SL.

In the laboratory tests, all isolates performed similarly against *A. orientalis* as against *P. japonica*, with both *S. scarabaei* isolates being by far more virulent than the other isolates. Against *A. orientalis*, but not *P. japonica*, *S. scarabaei* Ad was also more virulent than *S. scarabaei* SL, which was more obvious at lower IJ rates and at the first evaluation (7 DAT). Interestingly, despite high levels of mortality at 14 DAT already at 25 IJs/larva for *S. scarabaei* SL (75%), and even at 13 IJs/larva for *S. scarabaei* Ad (78%), no more than 95% of the larvae were killed at 400 IJs/larva. This suggests a high level of variability in susceptibility among individual grubs. The observed high level of virulence is similar to that observed for *S. scarabaei* AMK001 in previous studies (e.g., [6,7]), in which LC_50_ and LC_75_ values had been determined for 14 and 23 IJs/larva, respectively, for *A. orientalis*, and for 9 and 13 IJs/larva, respectively, for *P. japonica*. The high virulence of *S. scarabaei* was confirmed in the greenhouse test with *A. orientalis*, but unlike in the laboratory test, both fresh isolates provided the same level of efficacy. However, the efficacy in greenhouse tests of the fresh isolates in the current study was about 12% and 6% lower at 0.31 × 10^9^ IJs/ha and 0.63 × 10^9^ IJs/ha, respectively, than that observed in previous studies for *S. scarabaei* AMK001 [6,7].

*Steinernema cubanum* HF was less virulent than both *S. scarabaei* isolates against *A. orientalis* and *P. japonica* but caused 57% and 54% mortality of the two species, respectively, at 100 IJs/larva, and 81% and 89%, respectively, at 400 IJs/larva. Thus, it was more virulent than the *H. bacteriophora* GPS11 strain and the *H. zealandica* X1 strain in a previous study [7]. In that study, the LC_50_/LC_75_ values of the *H. bacteriophora* GPS11 strain were 285/1940 IJs/larva against *A. orientalis* and 145/312 IJs/larva against *P. japonica*, and the LC_50_/LC_75_ values of *H. zealandica* X1 strain were 280/744 IJs/larva against *A. orientalis* and 253/723 IJs/larva against *P. japonica*. A literature review did not reveal any virulence or efficacy observations on *S. cubanum*, even in a comprehensive book chapter about EPN studies in Cuba, where the species was originally isolated [19]. Moreover, the species had not been isolated outside of Cuba before we found it infecting white grubs at Shark River Golf Course in New Jersey, but it was also regularly found during a 3-year study in soil samples from two other golf courses in central New Jersey [20].

*Steinernema glaseri* SH was the least virulent EPN isolate against *A. orientalis* and *P. japonica*, causing no more than 43–57% mortality even at 400 IJs/larva and providing the lowest level of control against *A. orientalis* in the greenhouse experiment. In comparison with other *S. glaseri* strains, this level of virulence and efficacy was comparable against *A. orientalis* but lower against *P. japonica* [1,4].

None of the isolates showed high virulence against *C. borealis* third instars. However, against this species, *S. glaseri* was not significantly less virulent than the strain causing the highest mortality, *S. scarabaei* Ad, whereas *S. scarabaei* SL was the least virulent isolate. The observed virulence levels were similar to those previously observed for *S. scarabaei* AMK001, *H. zealandica* X1, and *H. bacteriophora* GPS11 [7], and *S. glaseri* strains [4].

One of the motivations to collect and test fresh EPN isolates from infected white grubs in the field was the suspected loss of virulence in the original isolate AMK001 of *S. scarabaei*. We had observed more limited mortality of *A. orientalis* and *P. japonica* larvae when exposed to *S. scarabaei* AMK001 for culture maintenance purposes. With an inoculum of about 50 IJs per larva in 30 mL cups with soil and grass, only about 50% of larvae were infected in more recent years before this study. Based on our previous studies [7,9], we expected closer to 90% infection at that rate. However, the current study showed that despite 19 years in laboratory culture, *S. scarabaei* AMK001 had maintained a high level of virulence, similar to that observed in previous studies [7,9] and similar to that of the fresh *S. scarabaei* Ad isolate. The lower levels of infection in more recent culturing events may have been caused by extended periods between culturing, with IJs sometimes having been held for as long as 12 months at 6 °C in tissue culture flasks before inoculation. After such a long storage time, *S. scarabaei* IJ are still >90% alive (A.M.K., personal observation) but may have lost some of their infectivity. *Steinernema scarabaei* IJs stored at 8 °C in tissue culture flasks for up to 18 weeks (4.2 months) did not show any reduction in infectivity [21], but it is likely that continued storage would eventually lead to loss in infectivity.

The preservation of the high level of virulence in *S. scarabaei* AMK001 over 19 years of laboratory culture is likely to be at least in part related to the use of natural host white grubs for rearing and the exposure being conducted in soil with grass. If the culture had been maintained using the common approach for EPN culture with waxworms as hosts exposed on filter paper in Petri dishes, its virulence may not have been preserved as well. We did not conduct field experiments with the new strains, but given that greenhouse pots with grass in soil fairly well represent field conditions for turfgrass, it is likely that similar results would be observed in turfgrass field trials for short-term white grub control. For long-term suppression, other factors may be also important that were not tested, in particular, IJ persistence and reproductive capacity in the hosts. However, based on empirical observations from rearing *S. scarabaei* in its natural hosts in the laboratory, all the *S. scarabaei* isolates used in this study seemed to produce similar numbers of IJs (A.M.K, personal observation), which are also similar to what *S. scarabaei* AMK001 produced soon after its original isolation [10].

Future research should examine the potential of the here-tested EPN strains for long-term suppression of white grubs in turfgrass and other systems. EPN isolates that are more closely adapted to key insect pests and have not lost their ability to persist well as IJs in the soil between infection cycles may not necessarily provide higher immediate control of the pests, but may suppress them sufficiently to prevent damage for extended periods of time with only periodical reapplications at low rates. Natural *S. cubanum* populations persisted throughout a 3-year study on golf courses, with populations peaking late in the season when third-instar white grubs were available as hosts [20]. Periodic broadcast applications at low rates of this or similarly well-adapted species may show great promise for long-term suppression of white grubs. Future research should also explore the benefits of culturing EPN isolates in their natural hosts, if known, for better preservation of their infectivity, virulence, and persistence in the natural environment.

## Figures and Tables

**Figure 1 insects-15-01022-f001:**
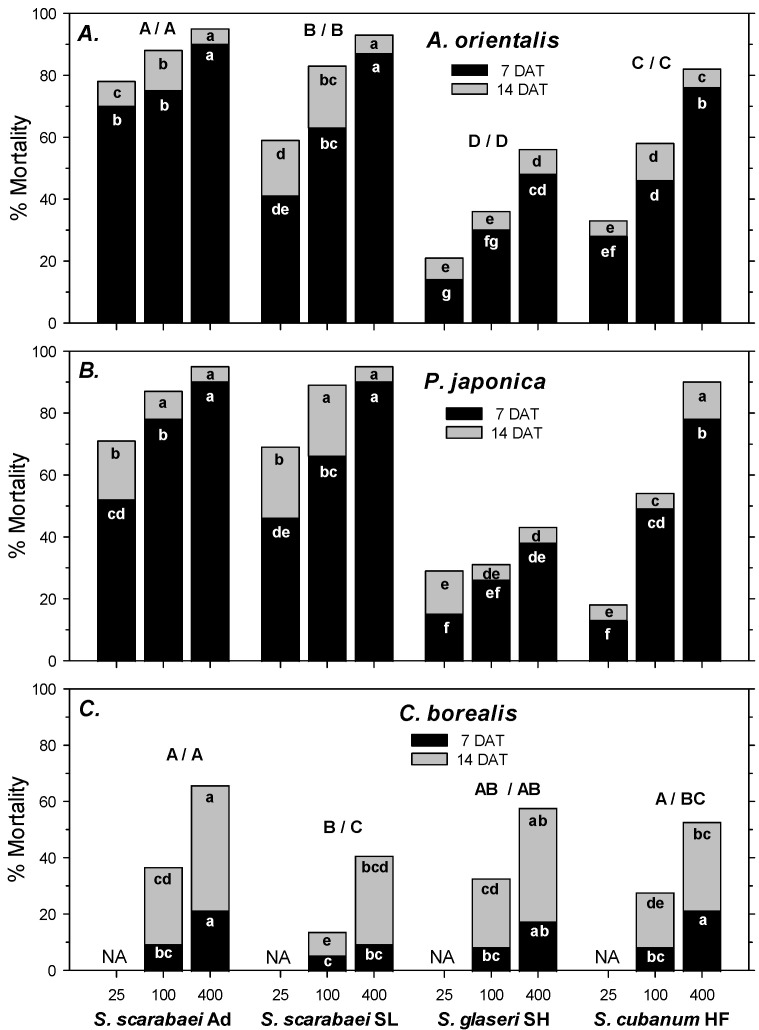
Percent mortality (control-corrected; mean) of third-instar *Anomala orientalis* (**A**), *Popillia japonica* (**B**), and *Cyclocephala borealis* (**C**) after treatment with the entomopathogenic nematode isolates *Steinernema scarabaei* Ad, *S. scarabaei* SL, *S. glaseri* SH, and *S. cubanum* HF (each at 25, 100, and 400 infective juveniles/larva) in 30 mL cups with soil and grass. Lower case letters in the black (grey) section of bars indicate significant differences among treatments at 7 (14) DAT; upper case letters above bars indicate significant differences among isolates across rates (7/14 DAT) (*p* < 0.05).

**Figure 2 insects-15-01022-f002:**
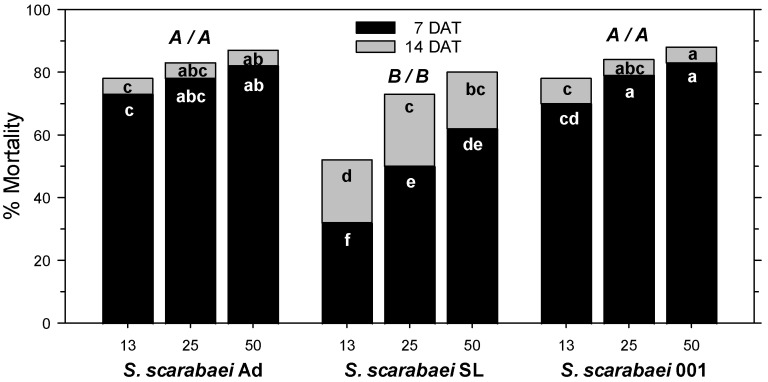
Percent mortality (control-corrected; mean) of third-instar *Anomala orientalis* after treatment with the entomopathogenic nematode isolates *Steinernema scarabaei* Ad, *S. scarabaei* SL, and *S. scarabaei* AMK001 (each at 13, 25, and 50 infective juveniles/larva) in 30 mL cups with soil and grass. Lower case letters in the black (grey) section of bars indicate significant differences among treatments at 7 (14) days after application (DAT); upper case letters above bars indicate significant differences among isolates across rates (7/14 DAT) (*p* < 0.05).

**Figure 3 insects-15-01022-f003:**
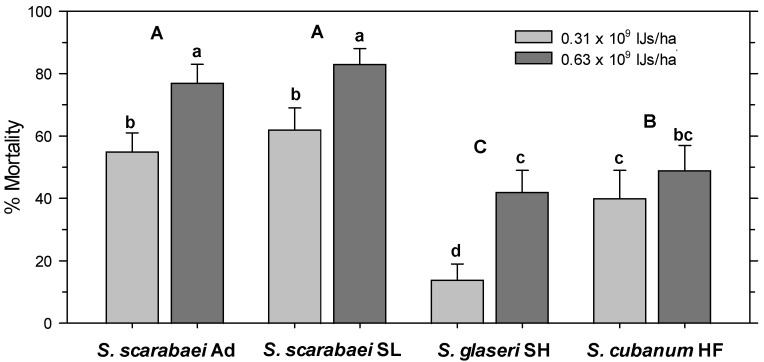
Percent mortality (control-corrected; mean) of third-instar *Anomala orientalis* after treatment with the entomopathogenic nematode isolates *Steinernema scarabaei* Ad, *S. scarabaei* SL, *S. glaseri* SH, and *S. cubanum* HF (each at 0.31 × 10^9^ and 0.63 × 10^9^ infective juveniles/ha) in 1-L pots with grass at 14 days after application. Lower (upper) case letters indicate significant differences among treatments (among isolates across rate) (*p* < 0.05).

## Data Availability

The data presented in this study are available upon request from the corresponding author.

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
