# Peer review of "New Strains of the Entomopathogenic Nematodes Steinernema scarabaei, S. glaseri, and S. cubanum for White Grub Management"

_insects, 2024, doi:10.3390/insects15121022_

Round 1

Reviewer 1 Report

Comments and Suggestions for Authors

This paper describes some experiments on investigating new strains of Steinernema scarabaei, S. glaseri, and S. cubanum for white grub management in laboratory and greenhouse conditions. I think the work is relatively novel and worthy of publication.

However, there are some questions, which need to be addressed before acceptance.

General considerations:

1.      In key words: “Insect-parasitic nematodes” required?

2.      In laboratory experiments authors mentioned that “Dry soil” is better to expand a little like the type of soil, soil compositions etc., and the greenhouse experiment mentioned “sandy loam” also soil composition is required.

3.      Is there any reason for assessing the larval mortality specifically at 7 and 14 DAT.?

4.      In the first two experiments, the author took three different concentrations of EPN against A. orientalis and P. japonica, but why in the third experiment, only two concentrations were used against C. borealis.

5.      On what basis the concentration of EPN (313 and 625 IJs/pot) was selected for the greenhouse experiment? Is it based on LC50 or Lc75?

6.      Why was there a difference in the number of pots per treatment in the first and second experiment run?

7.      On what basis A. orientalis selected for the greenhouse experiment as all the EPN isolates performed equally against A. orientalis and P. japonica?

8.      I feel it is better to suggest the software name in Statistical Analysis.

9.      In Statistical Analysis, the authors mentioned that “Means ± SEM are presented.” Is there any particular region to show SEM? Generally, researchers use standard deviation or Standard error.

10.  I suggest using any one strain or isolate throughout the MS. 

Author Response

Thank you for you edits and suggestions.  Please see the attached doc for response.

Reviewer 2 Report

Comments and Suggestions for Authors

A very well written manuscript relevant for the community. I tried to find anything to criticise (as I think this is the role of a caring reviewer) but did not succeed. Only point I could mention is that the number of self-citations is a bit high. Maybe you could add a few other references (if there are).

congrats to the authors

Author Response

(The authors gave the same response as above.)

Reviewer 3 Report

Comments and Suggestions for Authors

The authors conducted a laboratory and semi field assay to assess the recently isolated EPN species. The performance of EPNs species shows great variation depending on the application environment and isolates. Therefore, these tests must be conducted to identify the most virulent EPN species.

The manuscript is well-written and established. However, I have a few comments on the manuscript.

"As the soil and soil litter environment is their natural habitat, EPNs tend to be particularly promising for the control of soilborne insect pests." please add related references here.

Please, add a short table of used nematode species in the experiments with their location and Genbank accession numbers.

Figures may be confusing to readers, please check if there is any error.

 0.313 and 0.625 × 109 IJs/ha, why did the authors chose these concentrations? Any specific reason, please clarify.

Author Response

(The authors gave the same response as above.)

Reviewer 4 Report

Comments and Suggestions for Authors

In my understanding, research on data about potential pest control agents is always important. Very interesting, the fact that one of the isolates preserves its activity 19 years after it was isolated.

Author Response

(The authors gave the same response as above.)
